# Genome-Wide Scans and Transcriptomic Analyses Characterize Selective Changes as a Result of Chlorantraniliprole Resistance in *Plutella xylostella*

**DOI:** 10.3390/ijms232012245

**Published:** 2022-10-13

**Authors:** Wenting Dai, Bin Zhu, Marcel van Tuinen, Tao Zhu, Dongliang Shang, Pedro Almeida, Pei Liang, Hidayat Ullah, Liping Ban

**Affiliations:** 1Department of Grassland Resources and Ecology, College of Grassland Science and Technology, China Agricultural University, Beijing 100193, China or; 2College of Plant Protection, China Agricultural University, Beijing 100193, China or; 3Naturalis Biodiversity Center, Understanding Evolution Group, The Natural History Museum, 2333 CR Leiden, The Netherlands; 4College of Animal Science and Technology, China Agricultural University, Beijing 100193, China; 5Department of Genetics, Evolution & Environment, University College London, London WC1E 6BT, UK; 6Department of Agriculture, The University of Swabi, Anbar-Swabi 23561, Khyber Pakhtunkhwa, Pakistan or

**Keywords:** chlorantraniliprole resistance, sweep selection, MAPK signaling pathway, metabolic resistance, cytochrome P450 monooxygenase

## Abstract

Pesticide resistance in insects is an example of adaptive evolution occurring in pest species and is driven by the artificial introduction of pesticides. The diamondback moth (DBM), *Plutella xylostella* (Lepidoptera: Plutellidae), has evolved resistance to various insecticides. Understanding the genetic changes underpinning the resistance to pesticides is necessary for the implementation of pest control measures. We sequenced the genome of six resistant and six susceptible DBM individuals separately and inferred the genomic regions of greatest divergence between strains using F_ST_ and θ_π_. Among several genomic regions potentially related to insecticide resistance, *CYP6B6-like* was observed with significant divergence between the resistant and susceptible strains, with a missense mutation located near the substrate recognition site (SRS) and four SNPs in the promoter. To characterize the relative effects of directional selection via insecticide tolerance (‘strain’) as compared to acute exposure to insecticide (‘treatment’), four pairwise comparisons were carried out between libraries to determine the differentially expressed genes. Most resistance-related differentially expressed genes were identified from the comparison of the strains and enriched in pathways for exogenous detoxification including cytochrome P450 and the ABC transporter. Further confirmation came from the weighted gene co-expression network analysis, which indicated that genes in the significant module associated with chlorantraniliprole resistance were enriched in pathways for exogenous detoxification, and that *CYP6B6-like* represented a hub gene in the “darkred” module. Furthermore, RNAi knock-down of *CYP6B6-like* increases *P. xylostella* sensitivity to chlorantraniliprole. Our study thus provides a genetic foundation underlying selection for pesticide resistance and plausible mechanisms to explain fast evolved adaptation through genomic divergence and altered gene expression in insects.

## 1. Introduction

Pesticide resistance in insects represents an excellent genetic model to explain how insects adapt to the relatively short-term and strong selection pressure exerted by non-native chemicals. Understanding the genetic mechanisms underlying pesticide resistance is thus advantageous for pest control management. As a major pest of Brassica vegetable and oilseed crops throughout the world, the diamondback moth (DBM), *Plutella xylostella* L. (Lepidoptera: Plutellidae), is estimated to cause approximately US$0.77 billion management tool and crop losses annually in China alone [1]. Chemical insecticides remain the main management route to control the DBM. However, the abuse of various insecticides, coupled with a short generation time and the large overlapping of generations in DBM [2], have promoted DBM field resistance to all major kinds of insecticides, including organophosphates, carbamates, pyrethroids, and *Bacillus thuringiensis* (Bt) Cry toxins [3]. Chlorantraniliprole was the first commercialized anthranilic diamide insecticide widely used for pest control, especially for Lepidoptera pests [4]. However, within three years of its introduction in Guangdong in 2008, DBM had evolved a high level of resistance to chlorantraniliprole in the field [5]. In general, insecticide-resistance mechanisms have been described as three major types, (i) metabolic resistance that involves overexpression and elevated catalytic activity of detoxification enzymes; (ii) target resistance that involves mutation of the insecticide target site; and (iii) penetration resistance that involves modifications of the cuticle [6,7]. Previous research on the chlorantraniliprole-resistance mechanism in DBM mainly focused on the mutations in the target site, the ryanodine receptors (RyRs) [8,9,10,11]. More recently, the role of metabolism resistance in DBM that accompanies chlorantraniliprole resistance has been highlighted [12,13,14,15]. However, there has been no genome-wide scan of selection performed in *P. xylostella* to identify candidate chlorantraniliprole resistance-related genes.

The development of high-throughput sequencing technology, combined with the availability of genome-scale genetic data and statistical methods, is increasingly providing a general framework for genome-wide scans of selection (GWSS) to identify positively selected loci associated with several phenotypic traits. Methods based on population genetics’ statistics such as including F_ST_, π, iHS, Tajima’s D, XP-CLR, and XP-EHH have been widely used to seek out the genetic targets of artificial selection. Examples include genes associated with the domestic yak’s behavior and tameness [16], fat tail genes of Chinese indigenous sheep [17], meat and milk quality traits in cattle [18,19], and other economic traits of domestic animals [20,21,22]. In addition to domesticated traits, GWSS also has been used to identify adaptation to natural selective pressures, such as the adaptation of Dehong humped cattle to heat stress [23], Yanbian cattle to a cold climate [24], goats and sheep to a hot, arid environment [25], and Tibetan pigs to a high altitude [26]. There are also successful applications of GWSS to insecticide resistance, such as detecting genomic regions affected by DDT selection in *Drosophila melanogaster* [27] and genes associated with pyrthroid and DDT resistance in *Amyelois transitella* [28].

In this study, whole-genome resequencing was performed on two DBM strains (chlorantraniliprole-susceptible and -resistant strains), and F_ST_ for GWSS was used to detect genes with a signature of positive selection. The susceptible strain was maintained in laboratory conditions without exposure to any insecticide for 5 years, while the resistant strain originates from a chlorantraniliprole-susceptible population through six generations of chlorantraniliprole selection and backcrossing (Figure 1a). Secondly, transcriptional data were obtained, and analysis was performed to study the potential response to insecticide via altered gene expression of candidate genes. It is hypothesized that through these diverse approaches focusing on changes at the DNA and RNA level, many resistance-related genes may be detected. Among these candidates, *CYP6B6-like* was confirmed to be involved in chlorantraniliprole resistance by RNAi. Such knowledge will prove essential to the future management and use of insecticides if the genetic and biochemical mechanisms of insecticide resistance are better understood.

## 2. Results

### 2.1. Genome Resequencing and Genetic Variation

Genome resequencing of 6 resistance and 6 susceptible DBM individuals yielded 131.03 G of clean data. A total of 1.07 million high-quality SNPs were detected among all samples, and these SNPs were annotated using SnpEff (v4.3). Principal component analysis (PCA) divided the 12 DBM genome samples into two populations, matching the resistant and susceptible strains (Figure 1b). The first PC and the second PC explained 20.53% and 11.25% of the total variation separating the two populations, respectively. Reduced genetic diversity was observed among the resistant strain samples, by estimation of average θ_π_ (*P* < 2.22 × 10^−16^) in both strains (Figure 1c). RNA-Seq expression data came to the same result (Figure 1d, see below for details). These results implicated higher diversity in the susceptible strain than the resistant strain, whose tolerance to chlorantraniliprole differs by 401.4-fold (Table 1). This pattern is in accordance with the expectation of selection on the resistant strain by selecting only the most pesticide-tolerant insects in subsequent generations for the continuation of the strain.

### 2.2. Selective-Sweep Analysis Identifies Candidate Genes Associated with Insecticide Resistance

To detect signatures of positive selection associated with insecticide resistance, we searched the DBM genome for regions with high F_ST_ between susceptible strains and resistant strains (F_ST_ > 0.50), yielding 1501 genes. All 1501 genes were used for KEGG analysis and gained a total of 132 enrichment pathways (Appendix A), with 7 pathways significantly enriched associated with signal transduction (MAPK signaling pathway and Wnt signaling pathway), lipid metabolism (fatty acid degradation), and development and regeneration (cytoskeleton proteins) (Figure 2). Because insecticide resistance is known to be related to metabolic resistance, target resistance, and penetration resistance according to previous studies, we focused our analysis on detoxifying metabolic enzymes, insecticide receptors, cuticle proteins, ion channels, digestive enzymes, and the ubiquitinase system (Appendix A). In this reduced dataset containing 63 genes, the metabolic detoxifying enzymes were overrepresented, including 13 P450s, 8 ABC transporters, and 1 UDP-glucuronosyltransferase, followed by ion channels. In addition, 41 genes coding insecticide receptors, cuticle proteins, ion channels, digestive enzymes, and the ubiquitinase system were also detected as positively selected genes (PSGs) (Appendix A).

In order to reduce false positives, we then calculated the ratio of nucleotide diversity (θ_π_ (susceptible/resistant)). Using only those regions in the top 5% of both F_ST_ (>0.42) and top 5% of log_2_ (θ_π_ (susceptible/resistant)) (>3.10) as the threshold, we identified 208 genes (Figure 3). Among 63 PSGs (F_ST_ > 0.50, Appendix A), seven (*LOC105381077*, *LOC105395328*, *LOC105386242*, *LOC105381169*, *LOC105394494*, *LOC105381143*, and *LOC105385997*) were in the top 5% of the F_ST_ and log_2 (_θ_π_ ratio. In particular, *CYP6B6-like* (*LOC105395328*) showed both high F_ST_ (0.65) and θ_π_ ratio values (6.09) compared to neighboring regions, suggesting functional importance (Figure 4a). In resistant individuals of DBM (n = 6), we identified eight SNPs in the intragenic region of *CYP6B6-like*, including four intron variants (SNP48704, SNP48706, SNP48716, and SNP48729), one missense variant (SNP48968) and three synonymous variants (SNP48570, SNP48597, and SNP48999), which were absent in susceptible individuals (n = 6) (Figure 4b). These results indicate that these mutations of *CYP6B6-like* may be associated with insecticide resistance. Furthermore, the genomic sequence of *CYP6B6-like* was used as input in the DBM-DB blastn query (http://iae.fafu.edu.cn/DBM, accessed on 10 July 2020), and the result showed that *CYP6B6-like* was the gene *CYP6BG5* (Gene ID: Px014216) described in Yu et al. [29], which is over-expressed in chlorpyrifos- and fipronil-resistant strains of DBM.

To further analyze the effect of mutation on the CYP6B6-like protein, we focused on the missense variant (SNP48968). SNP48968 caused an amino acid mutation (A486V) at the protein level (Appendix A). Sequence alignments show that CYP6B6-like has five conserved motifs common to insect CYP6s: The WxxxR motif, the GxE/DTT/S motif, the ExLR motif, the PxxFxPE/DRF motif, and the PFxxGxRxCxG/A motif (Appendix A). The three-dimensional structure of CYP6B6-like (Appendix A) was then predicted using I-TASSER (C-score = 0.56, TM-score = 0.79 ± 0.09). The quality assessment of VERIFY, ProQ, and ModFOLD indicated that the resulting model is of good quality. The missense mutation A486V of CYP6B6-like is not in the conserved region, but in the highly variable region adjacent to the SRS6 (SRS: Substrate recognition site) (Appendix A).

### 2.3. Differentially Expressed Genes (DEGs) and KEGG Enrichment Analysis

To investigate the effects of acute insecticide exposure (‘treatment’) and directional selection via insecticide tolerance (‘strain’) on the gene expression level of DBM, four pairwise comparisons were carried out between libraries to determine the DEGs (RT vs. T, RCK vs. CK, RT vs. RCK, and T vs. CK, with samples described in Table 2). The normalized RNA-Seq expression data were used to perform principal component analysis, and PC1, PC2, and PC3 explained 91.53%, 3.83%, and 2.25% of the total variation, respectively (Figure 1d). Primarily, the 12 individuals were divided into two main groups, which matched the resistant and susceptible populations. Secondarily, for the susceptible strain of DBM, individuals also clustered by treatment, while individuals representing the resistant strain of DBM did not show significant clustering by treatment. For further interpretation, genes with |log_2_FoldChange| > 1.5 and *P*_adjust_ < 0.05 were recognized as differentially expressed. Our results showed that there were 518 (365 up-regulated and 153 down-regulated, CK vs. RCK), 615 (453 up-regulated and 162 down-regulated, T vs. RT), 52 (47 up-regulated and 5 down-regulated, RT vs. RCK), and 221 (94 up-regulated and 127 down-regulated, T vs. CK) DEGs by comparison of different strains or treatments (Appendix A). We identified 117 genes related to insecticide resistance from four pairwise comparisons based on previous research (Appendix A), and the top 10 significantly expressed genes are listed in Appendix A. Except for DEGs in the comparison of insecticide-treated susceptible DBM with control susceptible DBM, the majority involved up-regulation, including the cytochrome P450s, acetylcholinesterase, ion channels, and cuticle proteins. The overlap of DEGs between treatments and strains was analyzed and is displayed in Venn diagrams (Appendix A). Only three genes were differentially expressed (DE) after acute insecticide exposure both in the susceptible and the resistant strain, while 201 common genes were DE between strains with or without exposure to the insecticides. Coupled with the expression pattern of DE P450s, which showed an obvious expression difference between strains instead of treatments (Appendix A), these data indicate a stronger response to long-time directional selection for the susceptible strain when comparing exposure to non-exposure, which was consistent with the instability of insecticide-resistance that the resistant level of insects would gradually reduce once without pesticide pressure. Further analysis of the DEGs between strains by association with known KEGG pathways categorized DEGs into 33 groups (Figure 5), including pathways related to metabolic regulation, exogenous detoxification (cytochrome P450, ABC transporter), and immune response (Toll and Imd signaling).

To study whether the expression level of genes under positive selection (F_ST_ > 0.50) would be different between strains or between treatments, we investigated which genes with high F_ST_ were also listed as DEGs (Appendix A). In total, 7.3% of PSGs were also DE between treatment or between strains, with more PSGs DE between strains. Among these candidate genes, we identified eight pesticide-resistance related genes, including cytochrome P450s, ubiquitin-protein ligase, and chymotrypsin (*LOC105396609*, *LOC105395328*, *LOC105388375*, *LOC105382116*, *LOC105387005*, *LOC105397691*, *LOC105397997,* and *LOC105386494*).

### 2.4. Identification of Hub Genes in Significant Modules Related to Resistance

Transcript data from all 12 samples from two strains of DBM (susceptible and resistant) were used to construct the module that consists of functionally related genes of a similar expression profile, according to the weighted gene co-expression network analysis (WGCNA) algorithm (Figure 6a). In the present study, the power of β = 6 was selected as the soft threshold to ensure a scale-free network (Figure 6b), and 16 modules were identified (Appendix A). Of these modules, the module eigengenes (ME) of the “magenta” and “darkred” module exhibited the high correlation with pesticide resistance (magenta: r = 0.94, *P* = 6 × 10^−6^; darkred: r = −0.79, *P* = 0.002) (Figure 6c). Thus, the two modules were considered the most meaningful module of co-expressed genes related to resistance. The set of genes was then further explored for functional enrichment analysis and the identification of key genes. A total of 5383 genes in the two modules were enriched for particular exogenous detoxification pathways including cytochrome P450 and ABC transporter, again highlighting the importance of metabolic resistance in chlorantraniliprole resistance (Figure 6d). Module membership vs. gene significance (MM vs. GS) in the two modules is highly correlated (Appendix A), illustrating that genes highly significantly associated with resistance are often also the most important elements of the significant module. We detect hub genes with GS > 0.80 and MM > 0.80 and show the expression pattern of the top 10 hub genes in “magenta” (Figure 7a) and “darkred” modules (Figure 7b). All of these hub genes show significant expression differences between strains, while showing no change in expression to chlorantraniliprole-treatment (Figure 7). In particular, there are two P450s, *CYP6k1-like* (*LOC105386477*) and *CYP6B6-like* (*LOC105395328*), in the two groups of hub genes showing a high expression level in the resistant insects, suggesting their important function in insecticide resistance.

### 2.5. Knock-Down of CYP6B6-like Enhances the Sensibility to Chlorantraniliprole

Considering the strong selective signature, constitutive overexpression, and identification as a hub gene of *CYP6B6-like*, we focus the analysis on it. In addition to the coding region mutation as mentioned above, several SNPs were found in the 2000 nt regions upstream of *CYP6B6-like* (Figure 8a), and potential transcription factor binding sites at these regulation regions of *CYP6B6-like* were predicted using Jaspar (http://jaspar.genereg.net/, accessed on 20 September 2020). Interestingly, these SNPs landed at two predicted transcription factor binding sites, which were in the upstream fragment of the *CYP6B6-like* (NW_011952654.1:45630-47636 and NW_011952654.1:45641-45646). Furthermore, our RNA-seq data show a significantly higher expression level in the resistant DBM compared to the susceptible DBM (Figure 8b). These findings suggest that genetic divergence resulting from artificial selection for chlorantraniliprole resistance in DBM may influence the expression of *CYP6B6-like* to some extent. Therefore, to verify its expression level is indeed linked to chlorantraniliprole resistance, we next knocked down *CYP6B6-like* by feeding dsRNA to third-instar larvae. The expression level of *CYP6B6-like* decreased by 89% after 24 h feeding of dsCYP6B6 compared with the control (Figure 8c). We then used chlorantraniliprole (0.042 mg/L) to treat the RNAi individuals. The mortality rates of dsCYP6B6-feeding larvae were higher than those of the control group fed with the elution solution (Figure 8d and Appendix A) (Student *t* test, *P* < 0.01 in 72 h). Taking all evidence together, these results demonstrate that *CYP6B6-like* plays a critical role in chlorantraniliprole resistance.

## 3. Discussion

Insect resistance to pesticides is an urgent problem in integrated pest management. In the present study, we compared whole genome data of DBM from a chlorantraniliprole-resistant strain with a chlorantraniliprole-susceptible strain and identified (a) lower nucleotide diversity in the resistant strain, consistent with directional selection, and (b) several PSGs related to insecticide resistance using a population differentiation index (F_ST_). From this comparison, several detoxification enzyme genes, pesticide target genes, digestive enzyme genes, ubiquitin-protein ligase, and cuticle protein genes were found to be subject to positive selection. After further correction for false positives using the nucleotide diversity ratio (θ_π_ ratio, θ_π_ (susceptible/resistant)), *CYP6B6-like* (*LOC105395328*) was identified as showing among the highest differentiation between resistant strains and susceptible strains, indicating it was under strong selective pressure. Transcriptome results also indicated that expression of many resistance genes was upregulated in the resistant strain and DEGs were enriched in pathways related to the detoxification metabolism.

The KEGG pathway analysis of PSGs filtered with F_ST_ > 0.50 demonstrated that many candidate genes were enriched in the mitogen-activated protein kinase (MAPK) signaling pathway. The MAPKs regulate various cellular programs in response to extracellular signals, and have many endogenous and exogenous substrates [30]. The MAPK-signaling pathway is involved in the drug resistance of cancer cells [31,32,33], a crucial factor of which is the reactivation of the MAPK-signaling pathway [34,35]. Although studies on the role of the MAPK signaling pathway in insecticide resistance are few, some exist. For example, the MAPK p38 pathway has been implicated in insect defense against Bt Cry toxins [36]. Furthermore, another study indicates that the MAPK signaling pathway confers Bacillus thuringiensis Cry1Ac resistance by altering the expression of midgut ALP and ABCC genes [37]. Additionally, recent research has revealed the role of the MAPK-signaling pathway in the regulation of P450-mediated insecticide resistance [38]. Our findings further confirm the potential role of the MAPK signaling pathway in chlorantraniliprole insecticide resistance, and also specify P450-mediated insecticide resistance. Similarly, studies have indicated the role of the Wnt signaling pathway in the chemoresistance of numerous cancer types by regulating the expression of resistance genes [39,40,41]. More specific to insecticide resistance, studies have shown differential expression of genes associated with the Wnt signaling pathway in chlorantraniliprole- and flubendiamide-resistant DBM [15,42]. The present GWSS analysis provides valuable information at the genome level regarding the role of signal transduction in the development of chlorantraniliprole resistance, although biological functions of the pathways linked to insecticide resistance need to be confirmed experimentally and regulation mechanisms in resistant strains also need further elucidation. Our results from the transcriptomic analysis show that DEGs are enriched in multiple pathways associated with resistance, including the metabolism of xenobiotics by cytochrome P450, drug metabolism of other enzymes, drug metabolism of cytochrome P450, ABC transporters, and glutathione metabolism. Detoxification metabolism to exogenous toxic substances is the most commonly described mechanism of insecticide resistance [43,44]. Strong correlations between the detoxifying enzyme activity and chlorantraniliprole resistance have been confirmed [45], and several studies have indeed also observed their overexpression in resistant DBM and verified the role of some detoxifying enzymes in chlorantraniliprole resistance of DBM by RNAi [13,14,46]. These findings are consistent with our results of selective-sweep analysis and transcriptome analysis (Figure 5, Appendix A) by pointing to the same pathways involved in pesticide resistance.

Among these detoxifying enzymes above, the genetic variation and expression difference of P450s are particularly prominent in the present study. P450s play an important role in endogenous and exogenous substance metabolism, the latter including drugs, pesticides, and plant toxins [47]. P450s in DBM are distributed in four main groups including CYP2, CYP3, CYP4, and a mitochondrial cluster. Members of the CYP3, mainly CYP6s and CYP9s, are shown to participate in xenobiotics metabolism and insecticide resistance, and members of the CYP4 clan are known to be induced by xenobiotics [29]. Our study comes to similar conclusions; the positively selected P450s and differentially expressed P450s are mainly CYP6s, CYP9s, and CYP4s (Appendix A). The transcriptional up-regulation of P450 genes and metabolic capacities of P450 proteins were regarded as important mechanisms of enhanced metabolic detoxification to insecticide [48]. At the transcript level, both the P450 constitutive overexpression caused by mutations in promoter sequence and the inducible expression caused by mutations in trans-acting factors or by their signaling cascades have been specifically associated with insecticide resistance [49]. At the protein level, mutations in catalytic site residues (SRS1, SRS4, SRS5, and SRS6), substrate access channel residues (SRS2 and SRS3), proximal surface residues, and interacting partners may account for the enhanced metabolic capacities of P450s [48]. In DBM, the overexpression of *CYP6BG1* in resistant strains has been reported to be involved in chlorantraniliprole resistance [14]. In the present study, we not only also observed overexpression of *CYP6BG1* (named *CYP6k1-like*, *LOC105386477*) in resistant DBM but also identified it as the hub gene in a significant module of co-expression associated with resistance (Figure 7a). In other lepidoptera, such as *Chilo suppressalis* [50] and *Spodoptera exigua* [51], the constitutive overexpression of multiple P450s in chlorantraniliprole-resistance strains has been confirmed. In our study, the expression pattern of P450s also shows constitutive overexpression of P450s in resistant strains (Appendix A), and we suggest the potential role of these overexpressed P450s in resistance to chlorantraniliprole, because their overexpression has been reported in chlorpyrifos- and fiprinol-, and chlorantraniliprole-resistance in DBM. However, while acute chlorantraniliprole-treatment does not induce increased expression of P450s, it could be connected to the insect growth stage, insecticide exposure time, insecticide type, and insecticide concentration [52]. These enzymes may not be sensitive to the inducer or do not have enough time to control their biosynthesis and respond to insecticide.

Combined with the result of F_ST_ and θ_π_, we found that a P450 gene, *CYP6B6-like* (*LOC105395328*), was under strong positive selection and had a high genetic divergence between resistant and susceptible DBM strains. As a member of CYP6s, it may participate in the metabolic detoxication of insecticide. A previous study showed that *CYP6B6-like*, also named *CYP6BG5*, was transcriptionally up-regulated in the chlorpyrifos- and fiprinol-resistance strains compared to susceptible strains [29]. Although the target site of chlorantraniliprole is different from chlorpyrifos and fiprinol, cross-resistance in insects is a common phenomenon as metabolic mechanisms of resistance could confer broad-spectrum resistance [3]. Moreover, *CYP6B6-like* shares a high identity with *CYP6BG1*, which has confirmed involvement in chlorantraniliprole resistance in DBM. Interestingly, it was also predicted as one of the target genes of 11 lncRNAs linked to chlorantraniliprole resistance in DBM [53]. Taken together, these data may be regarded as evidence that *CYP6B6-like* is involved in the detoxification metabolism of chlorantraniliprole. Furthermore, we uncovered a missense mutation in the resistant strain, causing an amino acid mutation A486V, which is located in a highly variable region near the SRS6. SRSs are reported to be located in regions of extensive sequence variation, and even a single amino acid mutation in the SRS regions could affect metabolic capacity [48,54]. For example, a variant in SRS6 (A484V) of CYP6B3 has been reported to force substrates to bind closer to the reactive oxygen on heme [55]. However, catalytic properties are not only associated with SRSs. Three point mutations in CYP6A2, located between SRS4 and SRS5 (R335S and L336V) and before SRS6 (V476L), are reported to have a prominent role in DDT-insecticide resistance [56]. In our study, the missense mutation A486V is located before SRS6, but its potential role in chlorantraniliprole resistance needs further study.

Alongside the possible change of CYP6B6-like protein function in the resistant strain due to directional selection, we also observed its overexpression in resistant strains, which may be associated with mutations in the transcriptional regulatory region (2000 nt upstream region of gene). As mentioned above, the promoter variations of P450s are related to constitutive overexpression [48]. For example, some transposon insertions of P450s are reported to be associated with insecticide resistance in *Drosophila melanogaster* [57,58]. Here, several substitutions (SNP45630, SNP45634, SNP45635, and SNP45641) were observed to be located at two predicted transcription factor binding sites (NW_011952654.1:45630-45636 and NW_011952654.1:45641-45646) (Figure 8a). However, the specific role of these predicted transcription factors in the transcriptional regulation of P450s needs further confirmation (Appendix A).

Alongside several P450s, we also identified several other genes with strong signs of selective sweeps (Appendix A). This finding is not entirely surprising, since metabolism resistance is a complex physiological process involving a number of enzymes and transporters. P450s, esterases, and UDP–glycosyltransferases (UGTs) are three critical enzyme categories participating in the metabolism of xenobiotics and ABC transporters transfer and excrete their products [59,60]. In DBM, overexpression of *UGT2B17* has been reported to be involved in chlorantraniliprole resistance [13]. The contribution of ABC transporters to chloramphenicol resistance is rarely reported, but a recent study revealed that ABC transporters were involved in the transport of chlorantraniliprole [61]. Interestingly, the overexpression of P450s and ABC transporters is also known to change the thickness and composition of i”sect’cuticles [62,63,64], which may also be caused by the overexpression of cuticle proteins [65,66]. The insect cuticle is a critical determinant of insecticide resistance by reducing insecticide penetration, which has been reported in various insects [67,68,69], though studies on this aspect are fewer than those that support metabolism resistance. It also appears that proteasomes are associated with the response to insecticides, because they increased energy and amino acid production [70,71]. In addition, trypsin has been confirmed to catalyze the degradation of deltamethrin [72], which may lead to a deeper understanding of the role of proteasomes in insecticide resistance. These studies mentioned above are in agreement with the possible role of candidate genes found in our study in the development of chlorantraniliprole resistance.

In summary, we provide insights into the genomic basis of chlorantraniliprole resistance of DBM. The abundance of information from genome and transcriptome analysis could facilitate further investigations of the detailed resistance mechanisms of DBM against chlorantraniliprole.

## 4. Materials and Methods

### 4.1. Insect Sample and Treatment

The susceptible strain (DBM-S) was initially purchased from the Pilot-Scale Base of Bio-Pesticides, Institute of Zoology, Chinese Academy of Sciences in 2014, and was maintained in the laboratory for 5 years without exposure to any insecticide. The resistant strain (DBM-R) was collected in 2017 from a field at Fuluo, Guangzhou, China. Resistant insects used for sequencing were from a near-isogenic line (NIL) constructed with the backcross method (Figure 1a), which referred to Zhu et al. [73]. Specifically, the resistant males (DBM-R) were crossed with susceptible females (DBM-S), and their offspring were reared on fresh cabbage leaves with a discriminating dose of chlorantraniliprole, resulting in 40–70% mortality. The surviving larvae were reared on chlorantraniliprole cabbage leaves to pupae, and then surviving males were backcrossed with DBM-S females, and the same selection was performed for their offspring using a discriminating dose of chlorantraniliprole. The backcross and the selection process were repeated for six generations to obtain the NIL resistant to chlorantraniliprole (NIL-R).

The toxicity of chlorantraniliprole to DBM-S and NIL-R (LC_50_) was determined by the leaf dipping method. Cabbage leaves measuring 5 cm × 5 cm were immersed for 10 s in different concentrations of chlorantraniliprole diluted with distilled water. We tested 8 concentrations using two-fold serial dilutions. The range of tested concentrations for susceptible and resistant strains was 0.003125–0.4 mg and 0.78125–100 mg/L, respectively. The leaves were then air-dried for 0.5 h and then placed individually into a Petri dish with filter paper. A total of 20 third instars of different populations were then put into the Petri dishes (with three replications). We recorded the mortality of insects 96 h after treatment. Larvae that did not move when pushed gently with a brush were considered dead. LC_50_ of DBM-R and NIL-R populations were analyzed by probit analysis using POLOPLUS 2.0 (LeOra Software, Petaluma, CA, USA). The resistance ratio (RR) was determined as the ratio of the NIL-R LC_50_/DBM-S LC_50_. Finally, DBM-S and NIL-R were used for DNA extraction, where the whole body of larvae was a sample, with six samples for each strain.

Subsequently, DBM-S and NIL-R were divided into two groups. One group (treatment) was reared with cabbage leaves treated with an LC_50_ concentration of chlorantraniliprole, whereas the cabbage leaves in the other group (control) were treated with water. The whole body of surviving larvae was used for RNA extraction, with three replicates per treatment.

### 4.2. Sequencing and Library Preparation

Genomic DNA was extracted using TIANGEN Magnetic Universal Genomic DNA Kit. Sequencing libraries were generated using NEB Next^®^ Ultra DNA Library Prep Kit for Illumina^®^ (NEB, Ipswich, MA, USA) following the manufacturer’s recommendations and sequenced on an Illumina Hiseq 4000 platform. We generated 131.03 G of paired-end reads of 150 bp length (Appendix A).

RNA was extracted using the TRIzol (Ambion, Austin, TX, USA) extraction method. The mRNA Sequencing libraries were generated using the NEB Next^®^ Ultra™ RNA Library Prep Kit for Illumina^®^ (NEB, Ipswich, MA, USA) following the manufacturer’s recommendations, and sequenced on an Illumina NovaSeq 6000 platform. We generated 112.93 G of paired-end reads of 150 bp length (Appendix A).

### 4.3. Read Alignment and SNP Calling

Because the quality of the NGS data is very important for the downstream sequence analysis, we filtered low-quality reads using the NGS QC Tool kit (v2.3.3) with default parameters [74]. The clean data were aligned using BWA-MEM (v0.7.15) to the *P. xylostella* reference genome (RefSeq accession: GCF_000330985.1). Sequence Alignment/Map (SAM) format files were dealt with using Picard tools SortSam (v2.2.4) for sorting, outputting them as Binary sequence Alignment/Map (BAM) format files. Duplicate reads were removed from individual sample alignments using Picard tools MarkDuplicates (v2.2.4) [75].

Before SNP calling, the Genome Analysis ToolKit (v3.6), RealignerTargetCreator, and IndelRealigner were used for global realignment. SNPs were called using GATK UnifiedGenotyper with the parameters min_base_quality_score of 20, stand_call_conf of 30, and stand_emit_conf of 30. Then, GATK VariantFiltration was used to filter the unconfident variant sites, with the filter (a) QUAL < 30.0; (b) QD < 5.0; (c) FS > 60.0; (d) MQ < 40.0; (e) MQRankSum < −12.5; (f) ReadPosRankSum < −8.0.

### 4.4. Selective-Sweep Analysis

To detect PSGs related to insecticide resistance to chlorantraniliprole, we scanned the genome for regions with a population differentiation index (F_ST_) and nucleotide diversity (θ_π_) ratio. F_ST_ and nucleotide diversity (θ_π_) were calculated with VCFtools (v0.1.13) using a 5 kb window with a 1 kb step. The negative and missing F_ST_ values were discarded, because these values have no biological interpretation [76]. The θ_π_ ratio was calculated as θ_π_ (susceptible)/θ_π_ (resistant).

### 4.5. Sequence Alignment and 3D Modeling of CYP6B6-like

The multiple alignments were constructed with MEGA-X [77] and Jalview (v2.11.1.0) [78]. 3D models of CYP6B6-like were generated by I-TASSER [79,80,81] and model quality was assessed with Verify 3D, ProQ, and ModFOLD [82,83].

### 4.6. RNA Interference

The cDNA fragments of *CYP6B6-like* (*LOC105395328*) were amplified by RT-PCR using the gene-specific primers conjugated with 20 bases of the T7 RNA polymerase promoter listed in Appendix A. The PCR products were used as templates for double-stranded RNA (dsRNA) synthesis using a MEGAscript RNAi kit (Ambion Inc., Vilnius, Lithuania) following the manufacturer’s instructions. The synthesized dsRNA was dissolved in elution solution, quantified with a NanoDrop2000 (Thermo Scientific, Wilmington, DE, USA), and stored at −20 °C until use. Third-instars *P. xylostella* larvae were individually placed in a Petri dish to avoid cannibalism, starved for 20 h, and then fed with a 5 mm × 5 mm cabbage leaf disc. Then, 500 ng dsCYP6B6 (dissolved in elution solution, provided in the RNAi kit) or the same volume of elution solution only were distributed to the cabbage leaf discs. After 24 h, the larvae that had completely eaten the cabbage leaf discs were chosen for the subsequent experiments. A total of 10 larvae were randomly collected after another 24 h for total RNA isolation and qRT-PCR analysis of gene expression. The larvae fed with the elution solution were used as controls. The elution solution used in this study was used as a negative control because our preliminary experiments showed that the *CYP6B6-like* transcript level had no significant difference between being fed dsGFP and being fed the elution solution (Appendix A). The primers used for qRT-PCR are listed in Appendix A.

To evaluate the effect of RNAi of *CYP6B6-like* on the susceptibility of *P. xylostella* to chlorantraniliprole, the larvae that had eaten dsRNA were raised on cabbage leaf discs treated with the LC_50_ of chlorantraniliprole, and mortality was recorded at 24 h, 48 h, 72 h, and 96 h post-treatment. Approximately 20 dsRNA-feeding larvae were tested in every three replicates.

### 4.7. RNA-Seq, Data Processing, and Differentially Expressed Gene (DEG) Analysis

To further identify whether the potential selective-sweep regions (F_ST_ > 0.50) could also affect gene expression, we compared the gene expression between the susceptible and resistant strains of DBM. Three susceptible and three resistant DBM samples were used for RNA-seq. Sample information is listed in Table 2.

We used fastp (v0.20.0) to filter out the bad reads and cut adapters with the default parameters [84]. After building a HISAT2 index using hisat2-build, we mapped the clean reads to the DBM reference genome using HISAT2 (v2.1.0) [85], with the output files of the SAM format. Then, the SAM format files were imported to SAMtools for sorting and building index files, and the index of the fasta file of the reference genome was built with SAMtools (v1.9). We used the htseq-count tool to calculate the counts of the reads mapping to each gene [86], These count data were used to determine gene expression variation by analyzing each resistant sample to a susceptible sample with DESeq2 (v1.24.0) [87] in R (v3.6.1) [88]. Genes with log_2_FC > 1.5 and *P*_adjust_ < 0.05 were considered to be upregulated after insecticide treatment, while those with log_2_FC > −1.5 and *P*_adjust_ < 0.05 considered down-regulated genes. We plotted the volcano plots and a Venn diagram of differentially expressed genes related to insecticide resistance in DBM using ggplot2 (v3.2.1) and VennDiagram (v1.6.20) in R [89,90], respectively.

We used the clusterProfiler (v3.12.0) [91], an R package, to determine significant Kyoto Encyclopedia of Genes and Genomes (KEGG) pathways enriched within the DEG dataset using *P*_adjust_ < 0.05 as a threshold.

### 4.8. Weighted Gene Co-Expression Network Analysis and Identification of Hub Genes in Significant Module

WGCNA is a systems biology method for identifying patterns that have relevance among genes in microarray samples. In the WGCNA network, interesting gene modules related to sample traits and key genes can be identified. In this study, the expression data of 12 samples were log-transformed using log_2_ (x + 1) and then used for WGCNA. The weighted co-expression network was constructed in accordance with the protocol of the WGCNA package in R [92]. We chose the power of 6, which is the lowest power for which the scale-free topology fit index reaches 0.90. We merged highly similar modules (correlation > 0.75) and obtained 16 modules. To identify modules that were significantly associated with the trait (resistance), we used module eigengenes (MEs) as principal components to correlate them with external traits and look for the most significant module. Gene Significance (GS) is defined as the correlation between the gene and the trait, and module membership (MM) is a measure of intra-modular connectivity. We identified hub genes as those genes with both a high Gene Significance (GS > 0.80) for resistance as well as a high Module Membership (MM > 0.80) in significant modules.

## Figures and Tables

**Figure 1 ijms-23-12245-f001:**
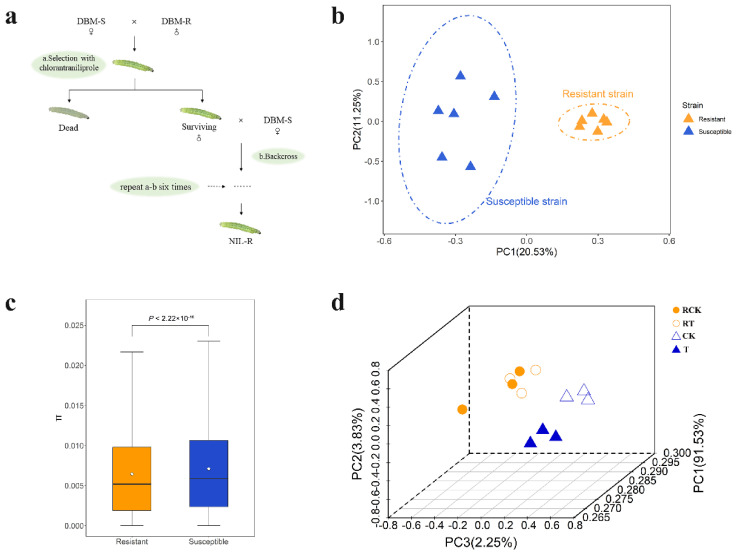
Genetic differentiation between susceptible and resistant strains of *Plutella xylostella*. (**a**) Construction of near-isogenic *Plutella xylostella* strains resistant to chlorantraniliprole. (**b**) Principal component analysis of SNPs. (**c**) Significant difference in population nucleotide diversity (θ_π_), as evaluated using the *t*-test (*P* < 2.22 × 10^−16^). (**d**) Principal component analysis of gene counts. Two colors represent two strains of *Plutella xylostella*, orange for the resistant strain and blue for the susceptible strain. Different graphs represent samples, RCK, Resistant stain without treatment by chlorantraniliprole (LC_50_); RT, Resistant stain after treated with chlorantraniliprole (LC_50_); CK, Susceptible strain without exposure to any insecticide; and T, Susceptible strain exposed to chlorantraniliprole (LC_50_).

**Figure 2 ijms-23-12245-f002:**
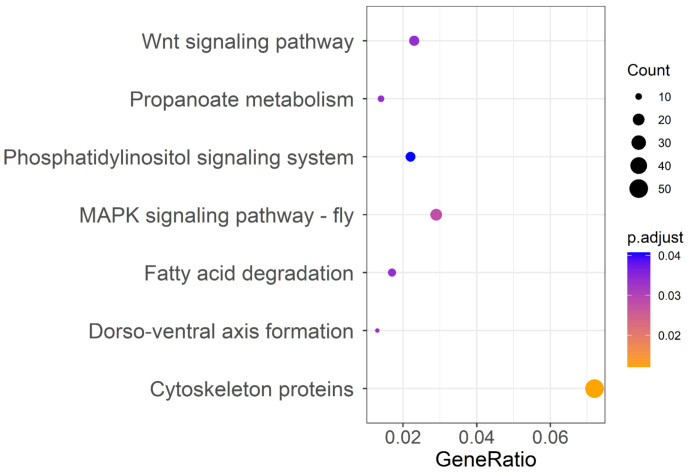
KEGG pathway annotation classification of genes located in F_ST_ > 0.50 region of the *Plutella xylostella* genome of strains differing in susceptibility to chlorantraniliprole. The ordinate left is the KEGG classification, and the abscissa is the ratio of gene enrichment in the pathway. The size of the round spot represents the number of genes enriched in the pathway, and the color represents the size of the *P*_adjust_ value.

**Figure 3 ijms-23-12245-f003:**
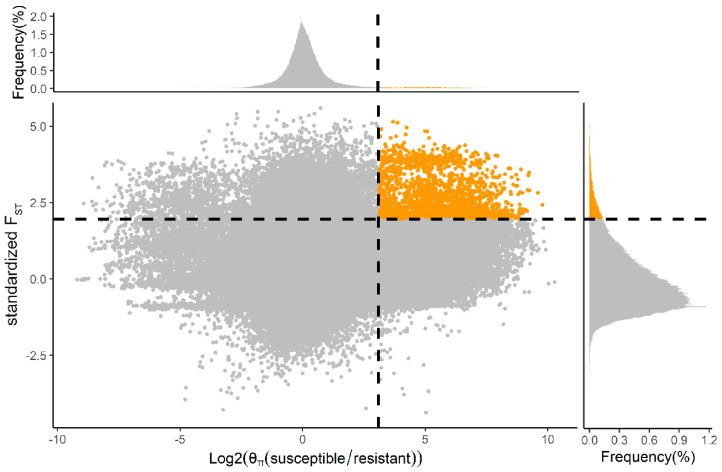
Distribution of F_ST_ and θ_π_ ratio (θ_π_ (susceptible/resistant)) across, respectively, the *X* and *Y* axes with associated frequency plots. The top-right corner where orange points are located represents the genome region under strong selection pressure for insecticide. The horizontal and vertical gray dashed lines represent the top 5% value of standardized F_ST_ (1.96, F_ST_ = 0.42) and log_2_ (θ_π_ (susceptible/resistant)) (3.10, θ_π_ (susceptible/resistant) = 8.56). The F_ST_ value in the figure is standardized using scale () function in R.

**Figure 4 ijms-23-12245-f004:**
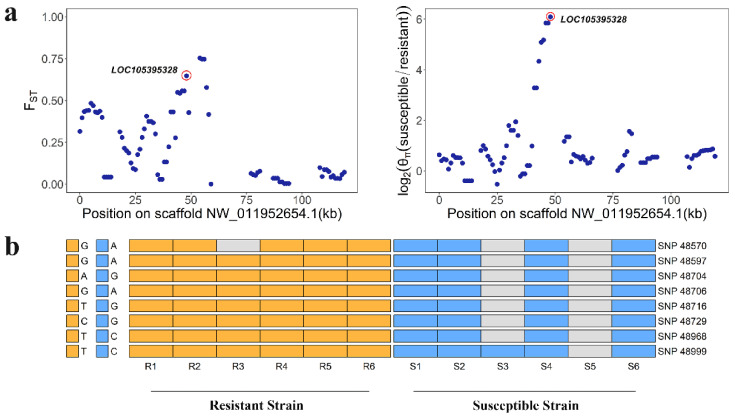
*CYP6B6-like* (*LOC105395328*) shows different genetic signatures between resistant-strain and susceptible-strain *Plutella xylostella*. (**a**) F_ST_ and log_2_ (θ_π_ (susceptible/resistant)) plot around the *CYP6B6-like* locus. The F_ST_ and log_2_ (θ_π_ (susceptible/resistant)) value of *CYP6B6-like* is almost highest for scaffold NW_011952654.1, circled in red. (**b**) The eight SNPs were identified in resistant strains of *Plutella xylostella* and absent in susceptible strains of *Plutella xylostella*. SNPs were named according to their position on the scaffold. The gray cell represents data missing.

**Figure 5 ijms-23-12245-f005:**
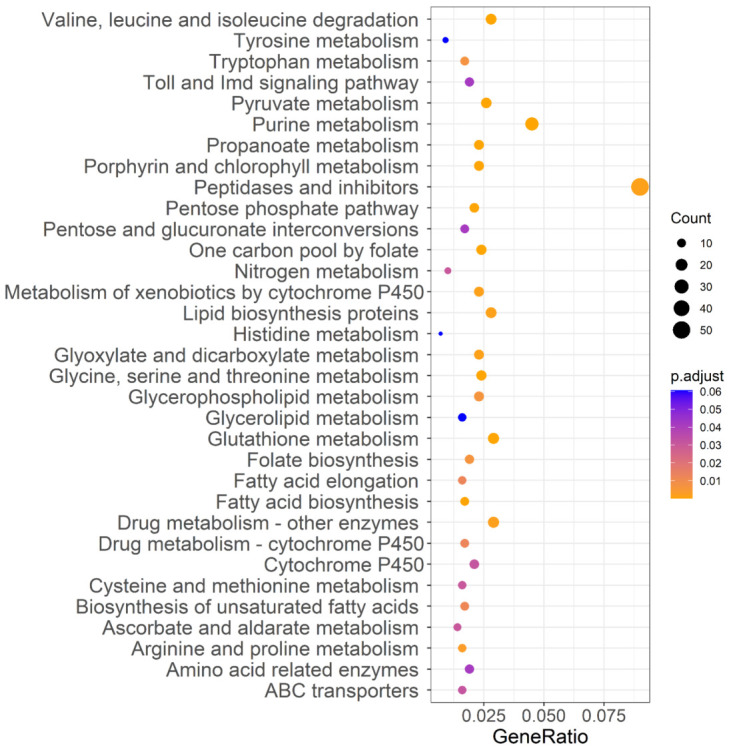
KEGG pathway annotation classification of DEGs between strains. The ordinate left is the KEGG pathway, and the abscissa is the ratio of genes enriched in the pathway. The size of the round spot represents the number of genes enriched in the pathway, and the color represents the size of *P*_adjust_ value.

**Figure 6 ijms-23-12245-f006:**
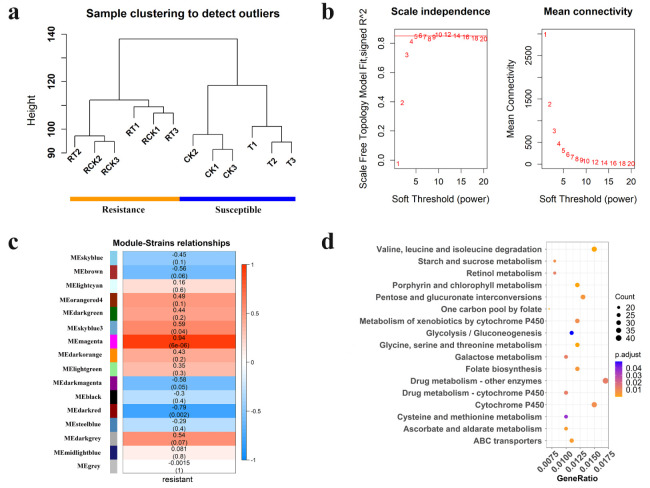
Weighted gene co-expression network analysis of 12 samples from two strains of *Plutella xylostella*. (**a**) Clustering dendrogram of 12 samples and the trait (resistance). RCK, Resistant stain without treatment by chlorantraniliprole (LC_50_); RT, Resistant stain after treated with chlorantraniliprole (LC_50_); CK, Susceptible strain without exposure to any insecticide; and T, Susceptible strain exposed to chlorantraniliprole (LC_50_). (**b**) Analysis of the scale-free fit index for various soft-thresholding powers (β). (**c**) Heatmap of the correlation between module eigengenes and chlorantraniliprole-resistance (the color from blue to red (−1→1) represents the relationship between module and trait). Each row corresponds to a module eigengene, and each column to a trait. Each cell contains the corresponding correlation and *p*-value. (**d**) KEGG pathway annotation classification of genes in “magenta” and “darkred” module. The ordinate left is the KEGG pathway, and the abscissa is the ratio of genes enriched in the pathway. The size of the round spot represents the number of genes enriched in the pathway, and the color represents the size of *P*_adjust_ value.

**Figure 7 ijms-23-12245-f007:**
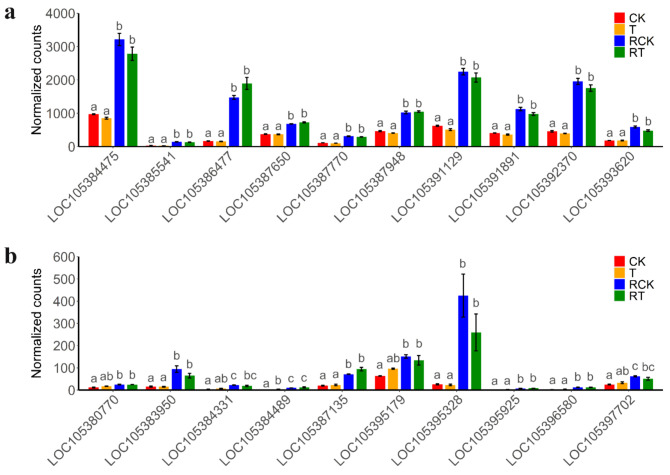
The expression pattern of hub genes in “magenta” (**a**) and “darkred” (**b**) modules. Bars with different colors indicated different strains and insecticide-treatment (shown in Table 2). Values sharing the different letters are significantly different at *P* < 0.05 (one-way ANOVA followed by Duncan’s multiple comparison tests, *P* < 0.05). LOC105392370: Pyrroline-5-carboxylate reductase; LOC105391891: Adenylosuccinate lyase; LOC105387948: Probable methylcrotonoyl-CoA carboxylase beta chain, mitochondrial; LOC105385541: Uncharacterized; LOC105384475: Trifunctional purine biosynthetic protein adenosine-3; LOC105391129: Trifunctional purine biosynthetic protein adenosine-3-like; LOC105387770: Growth-blocking peptide, long form-like; LOC105386477: Cytochrome P450 6k1-like; LOC105393620: Senecionine N-oxygenase-like; LOC105387650: Cytochrome b5; LOC105380770: Tudor domain-containing protein 1; LOC105383950: Scidic amino acid decarboxylase GADL1-like; LOC105384331: Uncharacterized; LOC105384489: Putative inorganic phosphate cotransporter; LOC105387135: Lactase-like protein; LOC105395179: Adrenodoxin-like; LOC105395328: Cytochrome P450 6B6-like; LOC105395925: Juvenile hormone epoxide hydrolase-like; LOC105396580: Uncharacterized; LOC105397702: Phosphoglucomutase-2-like.

**Figure 8 ijms-23-12245-f008:**
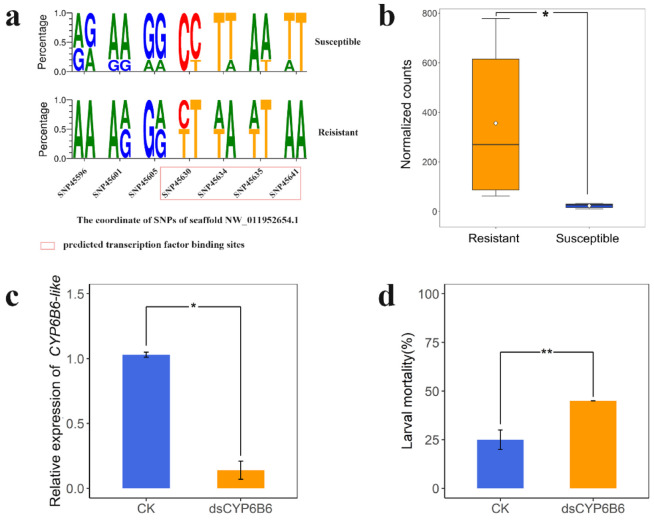
Functional analysis of *CYP6B6-like* (*LOC105395328*). (**a**) The SeqLogo plots show the diploid genotypes’ frequency of potential transcriptional regulatory region of *CYP6B6-like* (2000 nt regions upstream of gene), which were significantly different between resistant strains and the susceptible strain. (**b**) The expression levels of *CYP6B6-like* in susceptible and resistant strains. Normalized counts: Counts normalized with DEseq2. * indicates significant influences (Student’s *t* test, *P* = 0.026, n = 6). (**c**) The expression levels of the *CYP6B6-like* were significantly reduced after 24 h dsRNA feeding. * indicates significant influences (Student’s *t* test, *P* = 0.037, n = 3). CK indicates individuals fed with elution solution. Bars indicate ± SD. (**d**) The mortality rate of *Plutella xylostella* third instar larvae treated with LC_50_ concentrations of chlorantraniliprole (0.042 mg/L) after 72 h dsRNA feeding. ** indicates significant influences (Student’s *t* test, *P* = 0.0023, n = 3). Twenty individuals were used for each group. Bars indicate ± SD.

**Table 1 ijms-23-12245-t001:** Toxicity of chlorantraniliprole to two strains of *Plutella xylostella.*

Strain	LC_50_ (mg L^−1^) (95% CL)	Slope ± SE	χ^2^ (df) ^a^	RR ^b^ at LC_50_
Susceptible	0.042 (0.032–0.055)	3.35 ± 0.55	8.86 (8)	-
Resistant	16.86 (13.41–21.63)	2.04 ± 0.23	14.46 (12)	401.4

CL, confidence limit. ^a^ Chi-square value and degrees of freedom (df) as calculated by POLOPLUS. ^b^ Resistance ratio, equal to LC_50_ of resistant population/LC_50_ of susceptible population.

**Table 2 ijms-23-12245-t002:** Sample information of *Plutella xylostella* RNA-Seq.

Sample Name	Strain	Treatment
RT	Resistant	Exposed to chlorantraniliprole (LC_50_) for 24 h
RCK	Resistant	Without exposure to any insecticide
T	Susceptible	Exposed to chlorantraniliprole (LC_50_) for 24 h
CK	Susceptible	Without exposure to any insecticide

Note: We sequenced three replicates of each treatment.

## Data Availability

The 12 DBMs used in whole-genome resequencing analysis and the 12 DBMs used in RNA-seq analysis are accessible at the National Center for Biotechnology Information (NCBI) under Bio-Project accession numbers PRJNA664309. The whole-genome resequencing data and the transcriptome data have been deposited in SRA under SRR12674258-SRR12674281.

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
