# Peer review of "Genome-Wide Scans and Transcriptomic Analyses Characterize Selective Changes as a Result of Chlorantraniliprole Resistance in Plutella xylostella"

_ijms, 2022, doi:10.3390/ijms232012245_

Round 1
Reviewer 1 Report (Previous Reviewer 1)
Accept pending minor english grammar corrections, which can be done by the authors or the editorial office.
Author Response
Thanks very much for reviewer’s suggestion. And I tried to correct some grammar errors.
Reviewer 2 Report (Previous Reviewer 3)
There are a few comments for the authors to revise the manuscript.
1. In section 4.1, the authors may describe the concentrations with some details instead of “8-12 different concentrations, e.g. 0.5 mg/L, 1mg/L, etc.
2. The footnotes a and b should be present in the table as superscripts.
Author Response
Thanks very much for reviewer’s suggestion. The details of concentrations have been added in line 481-483. And superscripts also have been added in Table1.
This manuscript is a resubmission of an earlier submission. The following is a list of the peer review reports and author responses from that submission.
Round 1
Reviewer 1 Report
This is a well-designed study and can be recommended for publication pending some minor changes/ necessary justifications outlined below.
1. The LC50 table is lacking information on the total number of insects used for each LC determination. Likewise replication information for these bioassay and tested concentration range of chlorantraniliprole is not mentioned. This information should be included for transparency.
2. It is not clear why the authors did not use exogenous dsRNA control such as GFP/ green fluoroscent protein for the RNAi experiments. Authors should include justification for not using an exogenous dsRNA control. The exogenous control is necessary to determine if observed gene expression changes of the target gene are not caused by pleotropic effects of dsRNA.
3. Lastly, more description should be provided on how the authors determined the resistant individuals to be from near isogenic lines.
Reviewer 2 Report
The manuscript presented by Dai et al. reports a genomic and transcriptomic analysis performed on 2 populations of Plutella differing by their resistance or susceptibility to a chemical pesticide. This analysis reveal a P450 gene functional difference between the two lines. The involvement of this gene in pesticide resistance is confirmed by RNAi experiments.
As far as I can tell, the analyses have been conducted well and the conclusions are supported by the data. I can recommend the acceptance of this manuscript for publication.
Reviewer 3 Report
This study is focusing on the insecticide resistance mechanisms in diamondback moth. The authors conducted multiple experiments including RNA-seq and RNAi to identify the gene expression differences and single nucleotide substitution that caused chlorantraniliprole resistance. The research is scientific and the writing is fluent. Here are some questions and suggestions for authors to revise the manuscripts.
1. The authors used LC50 to determine the insecticide resistance level but no more description for the calculation of LC50. For instance, software or methods used.
2. Table 1 probably missed information about Chi-square and resistance ratio. There are footnotes but no data shown in Table 1.
3. Figure 1a described how the insecticide selection was conducted. However, the authors didn’t explain why both the male and female moths were treated with insecticide at first selection but backcrossed with non-selected males again. The authors may add more explanations.
4. Fig1d was described in section 2.3 and is present prior to figure 2. The authors may consider adjusting the order of figure 1d.
5. In the text, the authors used gene IDs. However, the format of gene IDs is not consistent. It should be revised to make all the gene IDs are in the same format. For instance, in line 152, the gene ids are numbers only.
6. The authors found 8 SNPs present in the intragenic region of CYP6B6-like and hypothesize the single substitutions involved in the resistance. However, how the 4 intron variants and 3 synonymous variants are involved in the resistance needs more clarification.
7. The findings indicate that “the missense variant SNP48968 did not cause structural damage to the protein”, how could the authors conclude that “it may cause the changing of binding affinity of CYP6B6-like to the insecticide”? If the authors have done the modeling and binding prediction study, it would be good evidence to support it.
8. In the methods, the authors stated “RNAi of CYP6B6-like…to chlorantraniliprole…mortality was recorded at 24h, 48h, 72h, and 96h post-treatment”, but in the results, the mortality rate of DBM larvae after insecticide treatment was collected after 72h dsRNA feeding. It should be explained and correct to make the story consistent.
9. How could the authors define the alive/dead insects? Need some more description.